# Humoral and T-Cell Mediated Response after the Third Dose of mRNA Vaccines in Patients with Systemic Lupus Erythematosus on Belimumab

**DOI:** 10.3390/jcm12031083

**Published:** 2023-01-30

**Authors:** Luca Quartuccio, Ginevra De Marchi, Rossana Domenis, Nicola Cabas, Silvia Guella, Antonella Paradiso, Cinzia Fabro, Antonio Paolo Beltrami, Salvatore De Vita, Francesco Curcio

**Affiliations:** 1Division of Rheumatology, Academic Hospital “Santa Maria della Misericordia”, ASUFC, Department of Medicine (DAME), University of Udine, 33100 Udine, Italy; 2Institute of Clinical Pathology, Academic Hospital “Santa Maria della Misericordia”, ASUFC, Department of Medicine (DAME), University of Udine, 33100 Udine, Italy

**Keywords:** lupus, belimumab, vaccine, coronavirus, COVID-19, immunity

## Abstract

Objective: To evaluate humoral and T-cell cellular-mediated immune response after three doses of SARS-CoV-2 mRNA vaccines in patients with systemic lupus erythematosus (SLE) under Belimumab. Patients and methods: 12 patients on Belimumab and 13 age-matched healthy volunteers were recruited. Patients were in remission or in low disease activity, and they were taking no corticosteroids or only low doses. None of the patients and controls had detectable anti-SARS-CoV-2 antibodies due to previous exposure to the virus. All the patients received three doses of mRNA anti-SARS-CoV-2 vaccines and the humoral and cellular-mediated response were tested 4 weeks after the second dose (T0), 6 months after the second dose (T1) and 4 weeks after the third dose (T2). Comparison with the control group was performed at time T0 (i.e., 4 weeks after the second dose). Total anti-SARS-CoV-2 RBD antibodies were analyzed using a diagnostic assay, while cellular-mediated response was evaluated using the interferon-gamma release assay (IGRA). Results: A humoral response was documented in all the patients at T0 (median 459; IQR 225.25–758.5), but the antibody titer significantly declined from T0 to T1 (median 44.7; IQR: 30.3–202; *p* = 0.0066). At T2, the antibody titer significantly increased from T1 (median 2500; IQR: 2500–2500), and it was not different from T0 (respectively *p* < 0.0001, *p* = 0.66). Cellular-mediated response significantly declined from T0 to T1 (*p* = 0.003) but not from T0 to T2 (*p* = 0.3). No differences were found between patients and controls at T0 as regards both humoral and cellular responses (*p* = 1.0 and *p* = 0.09 for humoral and cellular responses, respectively). Conclusion: The third dose of mRNA COVID-19 vaccine can restore both humoral and cellular immune response in SLE patients on Belimumab.

## 1. Introduction

*1.* 
*What is already known on this topic*
B-cell targeted therapies, in particular anti-CD20 monoclonal antibody, impair humoral response to mRNA vaccines against SARS-CoV-2, but few data are available for anti-BAFF antibody, which is the only licensed biologic treatment for SLE.


*2.* 
*What this study adds*
In patients exposed to Belimumab humoral response declines during the following 6 months after the second dose of mRNA vaccines, and some patients lose cellular-mediated response to mRNA vaccines against SARS-CoV-2 over time under Belimumab. After the third dose of mRNA vaccines both humoral and cellular-mediated response are restored under Belimumab.


*3.* 
*How this study might affect research, practice or policy*
Patients under Belimumab should receive further doses of mRNA vaccines to maintain both humoral and cellular-mediated response over time, and they should be monitored for the persistence of immune response over time.


SARS-CoV-2 vaccination with mRNA vaccine (mRNA-1273 e BNT 162b2) represented the cornerstone of the pandemic strategy to prevent severe complications of infection.

Immunosuppressive therapies potentially reduce both humoral and cellular-mediated response to vaccination, undermining its protective effect [1].

While many data have been accumulating in patients treated with Rituximab [2], still few data are available concerning patients with systemic lupus erythematosus (SLE) in Belimumab, which does not seem to impair the immunity against SARS-CoV-2 vaccines [3,4,5], even if the neutralizing capacity of anti-SARS-CoV-2 antibodies might be less efficient [3]. 

In this paper we present the full trend of B-cell and T-cell response after immunization with three doses of anti-SARS-Cov-2 vaccine (mRNA-1273 and BNT 162b2) in patients affected by SLE under stable treatment with Belimumab.

## 2. Patients and Methods

Between March and December 2021, all consecutive patients with adult SLE diagnosed according to ACR 1997 criteria, who were receiving Belimumab and completed the three doses vaccine schedule, were recruited. As controls, age-matched healthy volunteers, in at least 1:1 ratio with patients, selected among healthcare personnel of the Laboratory of Clinical Pathology, who received first 2 doses of BNT162b2 vaccine, were included. Humoral and cellular-mediated responses 4 weeks after the second dose (T0—2D4W), 6 months after the second dose (T1—2D6M) and, finally, 4 weeks after the third dose (T2—3D4W) were evaluated in the patients. In controls, we evaluated the vaccine response 4 weeks after the second dose (T0—2D4W).

Anti-SARS-CoV-2 antibodies were measured using two commercially available serological tests: iFlash-SARS-CoV-2 (Yhlo, distributed in Italy by Pantec) and Elecsys anti-SARS-CoV-2 electrochemiluminescence immunoassay analyzer (ECLIA) by Roche Diagnostics (Mannheim, Germany). The first is a paramagnetic particle-based chemiluminescence immunoassay used to assess anti-nucleocapsid (N) and anti-spike (S) IgG and IgM antibodies; it cannot recognize the SARS-CoV-2 protein RBD and is used to detect previous exposure to SARS-CoV-2 (cut-off, >10 U/mL). 

The second test is an immunoassay for quantitative determination of total (IgG/IgM/IgA) antibodies against SARS-CoV-2 spike 1 protein RBD and it is used to assess the serological response to vaccination (cut-off, >0.79 U/mL).

The T-cell-mediated cellular immune response to SARS-CoV-2 was evaluated using the interferon-gamma release assay (IGRA, Euroimmun). Freshly collected lithium-heparin blood (0.5 mL) was added to a stimulation tube coated with a peptide pool based on the S1 domain of the SARS-CoV-2 spike protein (Wuhan strain). Negative (blank) and positive (mitogen) controls were used to define individual background cell activation and to test the viability and responsiveness of T cells. The test results are considered reliable when the IFNγ released by the blank stimulation tube is <9 pg/mL and by the mitogen stimulation tube is >141 pg/mL. If this condition is not satisfied, the test is defined as indeterminate. The Coviferon test result is positive when the difference between the IFNγ released by spike stimulation tube and blank stimulation tube is >12 pg/mL.

Friedman test followed by Dunn’s test for repeated measures nonparametric ANOVA was applied to the patients’ group. *P* value below 0.05 was considered significant. Mann Whitney U test was applied to compare the humoral and T-cell-mediated cellular responses between patients and controls.

The study was approved by the local Ethics Committee, namely “Comitato Etico Unico Regionale” (CEUR) (approval number CEUR-2019-OS-233), and all the patients and volunteers provided their informed consent. 

## 3. Results

Twelve female patients with a median age of 47.6 ± 8 years in treatment with Belimumab (9/12 weekly subcutaneous; 3/12 monthly intravenous) were studied, as shown in Table 1, and 13 controls (8 females) with a median age of 46 ± 9 years (described in the Appendix A). Concomitant treatments are reported in Table 1. The ongoing immunosuppressive and corticosteroid treatments were left unchanged for the entire period under investigation.

The interval between the first and the second dose of vaccine was 23 ± 3 days, while between the second and the third dose was 188 ± 29 days.

None of the patients or controls had detectable anti-SARS-CoV-2 antibodies caused by prior exposure to the virus.

A humoral response was documented in all the patients at T0—2D4W, without difference compared to controls (median 459 U/mL; IQR 225.25–758.5 U/mL versus median 1133 U/mL; IQR 692.5–2260.5 U/mL, *p* = 1.0). At T1—2D6M, the antibody titer significantly declined (median 44.7 U/mL; IQR: 30.3–202 U/mL) as respect to T0—2D4W (*p* = 0.0066). At T2—3D4W, the antibody titer significantly increased (median 2500 U/mL; IQR: 2500–2500 U/mL), from T1 (*p* < 0.0001) and was not different from T0—2D4W (*p* = 0.66) (Figure 1, panel A). Moreover, the antibody titer at T2—3D4W was comparable to that reached by controls at T0—2D4W (median 2500 U/mL; IQR: 2500–2500 U/mL versus median 1133 U/mL; IQR 692.5–2260.5 U/mL, *p* = 0.24). 

At T0—2D4W, a T-cell-mediated cellular response was detectable in all the patients (median 172.79 pg/mL; IQR 59.94–271.99 pg/mL). At T1—2D6M, 9/12 patients still presented a T-cell-mediated cellular response, but it was quantitatively inferior than that observed at T0—2D4W (median 35.38 pg/mL; IQR 6.37–128.08 pg/mL; *p* = 0.0033) (Figure 1, panel B). One patient presented an indeterminate test and two patients had no longer a detectable T-cell-mediated cellular response. At T2—3D4W, a T-cell-mediated cellular response was found in 10/12 patients (median 103.99 pg/mL; IQR 19.57–272.64 pg/mL); in those patients with a previous indeterminate (1/3) or negative (2/3) response, we could document a positive test in 2/3 cases.

No statistical difference on the level of cellular response was found comparing T2—3D4W with T1—2D6M (*p* = 0.3) or T2—3D4W with T0—2D4W (*p* = 0.3). Overall, we observed a significant reduction of the T-cell-mediated cellular response at T1—2D6M, partially recovered after the third dose (T2—3D4W) (Figure 1, panel B). Again, no difference was found between patients and controls in both comparisons at T0—2D4W and T2—3D4W (T0—2D4W for patients: median 172.79 pg/mL; IQR 59.94–271.99 pg/mL vs. T0—2D4W for controls: 204.8 pg/mL; IQR 96.76–590.7 pg/mL, and T2—3D4W for patients: median 103.99 pg/mL; IQR 19.57–272.64 pg/mL vs T0—2D4W for controls; *p* = 0.09 and *p* = 0.2, respectively).

During the follow-up, 4 out of 13 patients got COVID-19 infection (Table 1, patient number 4, 5, 9, 12) after the third dose of vaccine, with a median time of 2.25 ± 0.95 months (in all cases within 4 months). All patients were taking Belimumab, three were taking HCQ, one patient was also taking MMF; no one was taking corticosteroids. Notably, 1 out of 4 patients who got COVID-19 infection, presented an undetermined cellular-mediated response at T2—3D6M.

All four patients were promptly referred to the Infectious Disease Department; one of them was treated with antiviral therapy, two with anti-SARS-CoV-2 monoclonal antibodies, while the last one did not receive any treatment. None of these patients needed hospitalization.

## 4. Discussion

This study described the trend of humoral and T-cell-mediated cellular response after three doses of anti-SARS-CoV-2 vaccine in SLE patients on Belimumab. Even if the low number of patients cannot establish definitive conclusions, we found that the third dose can restore the humoral response after 6 months from the second dose in this clinical setting. This observation is quite different to what reported for patients treated with Rituximab, who did not show a significant increase of the humoral response after the third dose of the vaccine [6], and even after the fourth dose with inactivated vaccine against SARS-CoV-2 [7]. Moreover, our results are consistent with those published very recently by Schiavoni et al. in an independent monocentric Italian cohort of 12 SLE patients treated with Belimumab [8]. In addition, the comparison with a control group of healthy people provided by our study, even if restricted to the T0, showed a similar vaccine response in SLE Belimumab-exposed patients.

Globally, humoral response in SLE patients under Belimumab has been reported as preserved in a high fraction of the patients [4,9,10], even if Belimumab has been associated with a reduced humoral response to inactivated vaccine against SARS-CoV-2 [5]. 

Conversely, the cellular-mediated response did not show substantial changes over time, even if in some patients it significantly decreased at the 6th month after the second dose similarly to what observed in other populations of immunosuppressed patients and in healthcare volunteers [10,11]. Overall, the third dose of the vaccine allowed to maintain the response, even if not in all the patients. The role of the concomitant treatments on this trend cannot be excluded [12,13], and longer clinical follow-up together with laboratory assessment of these patients is required. In fact, two of the three patients that had presented undetermined or absent cellular response (T1—2D6M) were using another concomitant immunosuppressant (methotrexate or tacrolimus). 

Overall, our results confirm the concept that in immunocompromised patients the third vaccine dose is not a booster dose, but it is a necessary completion of a regular vaccination cycle [14]. Indeed, the significant decline of the antibody titer after 6 months from the second dose and the incomplete T-cell-mediated cellular response after the third dose in a percentage of patients support the potential role of a fourth dose as a booster dose for those patients exposed to B-cell targeted therapies, like Belimumab or Rituximab [15]. Anyway, the favorable outcome observed in patients who got COVID-19 after the third vaccine dose could support the protective effect of the vaccine against the most severe outcomes, including hospitalization and death [16]. 

In conclusion, the third dose of mRNA COVID-19 vaccine restores the immune response in patients undergoing Belimumab, and this evidence supports the 3-dose regimen as the standard immunization among immunodeficient patients. The usefulness of monitoring the trend of humoral and T-cell-mediated cellular response in particular groups of immunocompromised patients has been also highlighted. The main limit of our study is the lack of data on the decline of the humoral and cell-mediated immune response after the 3rd dose, which could suggest the right timing for the fourth dose booster. As 4th dose is often recommended for heavily immunocompromised patients, it is clinically relevant to have data on the antibody and IGRA response after the fourth dose in patients receiving Belimumab.

## Figures and Tables

**Figure 1 jcm-12-01083-f001:**
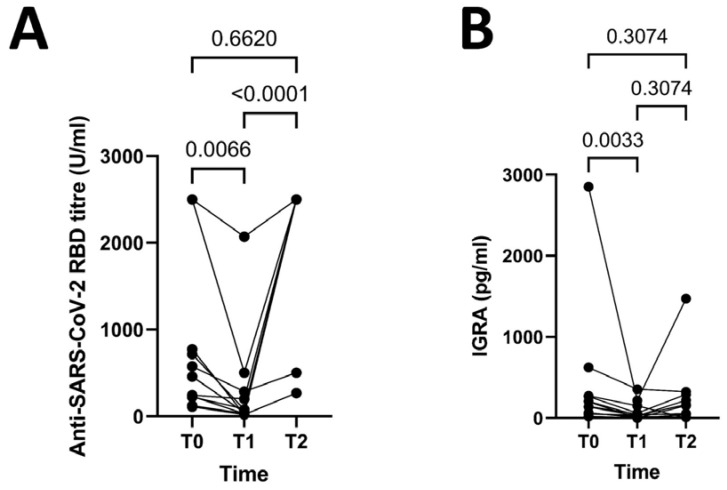
(**A**) Humoral response (by Anti-SARS-CoV-2 RBD antibody titre) at points T0 (4 weeks after the second dose), T1 (6 months after the second dose) and T2 (4 weeks after the third dose). (**B**) T-cell mediated response (by IGRA test) at points T0 (4 weeks after the second dose), T1 (6 months after the second dose) and T2 (4 weeks after the third dose).

**Table 1 jcm-12-01083-t001:** Characteristics and immune response of the patients.

	Patient 1	Patient 2	Patient 3	Patient 4	Patient 5	Patient 6	Patient 7	Patient 8	Patient 9	Patient 10	Patient 11	Patient 12
**Sex**	Female	Female	Female	Female	Female	Female	Female	Female	Female	Female	Female	Female
**Age**	58	46	59	42	44	61	51	49	40	42	40	40
**Disease**	SLE	SLE	SLE	SLE	SLE	SLE	SLE	SLE + APS	SLE+ APS	SLE	SLE + APS	SLE
**Concomitant Immunosuppressant**	No	No	MMF 1 g/day	No	MMF 500 mg/day	No	MTX 7.5 mg/week	AZA 100 mg/day	No	No	TAC 3 mg/day	CyA 150 mg/day
**Hydrxychloroquine**	Yes	Yes	Yes	Yes	Yes	Yes	Yes	No	Yes	Yes	Yes	No
**Corticosteroids**	No	No	PD 2.5 mg, 4 times/w	No	No	No	No	No	No	No	No	No
**Time beetween II and III dose of vaccine (days)**	177	188	174	136	213	180	210	142	232	231	203	173
**Remission (LLDAS)**	Yes	Yes	Yes	Yes	Yes	Yes	Yes	Yes	Yes	Yes	Yes	Yes
**Humoral and cellular response after the II dose (T0—2D4W)**												
Anti-RBD antibody titre, U/mL	715	2500	773	110	458	232	2500	223	460	243	124	576
Cellular response (quantitative), pg/mL	138	2851	144	48	58	624	21	207	265	65	202	274
Cellular response (qualitative)	Present	Present	Present	Present	Present	Present	Present	Present	Present	Present	Present	Present
**Humoral and cellular response 6-month after the II dose (T1—2D6M)**												
Anti-RBD antibody titre, U/mL	78	2072	21	21	45	30	504	86	43	202	33	NA
Cellular response (quantitative), pg/mL	26	216	49	38	18	356	2	33	61	3	2	150
Cellular response (qualitative)	Present	Present	Present	Present	Present	Present	**Indeterminate**	Present	Present	**Absent**	**Absent**	Present
**Humoral and cellular response after the III dose (T2—3D4W)**												
Anti-RBD antibody titre, U/mL	2500	2500	268	2500	2500	2500	2500	2500	2500	2500	2500	504
Cellular response (quantitative), pg/mL	218	1472	44	55	12	324	42	167	291	8	153	2
Cellular response (qualitative)	Present	Present	Present	Present	Present	Present	Present	Present	Present	**Indeterminate**	Present	**Indeterminate**
**COVID-19 infection after the third** **vaccine dose**	No	No	No	**Yes**	**Yes**	No	No	No	**Yes**	No	No	**Yes**

Legend: SLE: Systemic Lupus Erythematosus; APS: Antiphospholipid syndrome; MMF: Mycophenolate mofetil; MTX: Methotrexate; AZA: Azathioprine; TAC: Tacrolimus; CyA: Cyclosporine; PD: Prednisone; NA: not available.

## Data Availability

Data available on request due to privacy restrictions.

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
