# Peer review of "Humoral and T-Cell Mediated Response after the Third Dose of mRNA Vaccines in Patients with Systemic Lupus Erythematosus on Belimumab"

_jcm, 2023, doi:10.3390/jcm12031083_

Round 1
Reviewer 1 Report
A number of 12 patients treated with Belimumab, is a very low number, as far as I know, to establish any significant conclusions, upon the evaluation of humoral and T-cell cellular-mediated immune response after three doses of SARS-CoV-2 mRNA vaccines.
Not very clear how were the patients controls recruited.
No enough data are displayed about the needs of the third dose of mRNA COVID-19 vaccine to restore both humoral and cellular immune response in the Belimumab treated patients.
Reviewer 2 Report
The manuscript entitled: “Humoral and T-cell mediated response after the third dose of mRNA vaccines in patients with systemic lupus erythematosus on Belimumab” by Quartuccio et al. presented a comparative study between SLE patients under Belimumab and a control group assessing B and T-cell cellular-mediated immune response after three doses of SARS-CoV-2 mRNA vaccines. The work is very interesting, and it shows very important information about SLE patients’ vaccination, but I have some questions concerning the manuscript:
#1: I suggest that the control group data are presented in the abstract section.
#2: The authors should specify when the Friedman test followed by Dunn’s multiple comparisons test were used in the ‘Patients and methods’ section.
#3: Two of the three patients that had presented undetermined or absent cellular response (T1 – 2D6M) were using another immunosuppressant (MTX and TCA3). Could these therapies influence these results?
#4: Control group results must be clearly presented (To include in table may facilitate).
#5: What the conclusion about the SLE patients results (humoral and cellular responses) when compared to control group?
#6: In Figure 1 the different patients and times should be represented by different colors or symbols to facilitate the visualization.
#7 Which patients developed COVID-19 infection after the third dose of the vaccine? Since the study had a few patients, I think that it is better to identify each outcome regarding each patient.
#8 Are there differences between anti-body levels from control group and SLE patients at T1 – 2D6M time?
#9 The authors agree that “in immunocompromised patients the third vaccine dose is not a booster dose, but it is a necessary completion of a regular vaccination cycle”. However, is this also true for health control group?
